# A System Dynamics Approach for Improved Management of the Indian Mackerel Fishery in Peninsular Malaysia

Illisriyani Ismail [1,*], Pierre Failler [2], Antaya March [2] and Andy Thorpe [2]

1. International Institute of Aquaculture & Aquatic Sciences, Universiti Putra Malaysia, Port Dickson 70150, Malaysia
2. Blue Governance Centre, University of Portsmouth, Portsmouth PO1 3DE, UK
* Correspondence: illisriyani@upm.edu.my

**Abstract:** This paper presents the results of applying the system dynamics model to the Indian mackerel fishery (IM) in Malaysia. The main objective of this paper is to explore a more holistic approach to modelling options for the difficult task of managing an open access fishery. To this end, a system dynamics model was used to provide a general framework that incorporates several interacting factors that influence the management of the fishery. Such a model, which combines both biological and economic data, is appropriate for fisheries management practise. In this case, three policy scenarios (in addition to a business-as-usual scenario) based on a reduction in the number of boats fishing are used to simulate the behaviour of the stock IM. The results show that the optimal CPUE level is achieved with a 25% reduction in the total number of boats. In this scenario, a biomass of 127,432 metric tonnes IM of adult fish is predicted in 2050, with a catch of 32,884 tonnes for 2698 boats. In comparison, the biomass of IM in 2016 is 112,384 tonnes with a catch of 32,454 tonnes for 4616 boats. Based on these results, we can determine for the first time the optimal level of fishing capacity to ensure the sustainability of the fishery at IM. Overall, this work introduces a new method for simulating Malaysian fisheries data and new modelling methods that are not widely used in the Malaysian fisheries modelling field. The model benefits from current management and data availability.

**Keywords:** catch; mackerel; Malaysia; stock; system dynamics

## 1. Introduction

The Indian mackerel (IM), Rastrelliger kanagurta, is widely distributed in the Indian and Western Pacific Oceans and surrounding seas [1]. It is a migratory species, but little information is available on its migrations, behaviour, and spawning grounds [1,2]. In Peninsular Malaysia, the abundance of IM is associated with the abundance of phytoplankton in the water [3–5]. One of the most important indicators of phytoplankton abundance is the level of chlorophyll-a (Chl-a) in the water, which is relatively high in the region [6]. IM is considered one of the 'people's fish' due to the high demand from the Malaysian population [7], with an estimated annual consumption of $\geq 2$ kg/person. From 1980 to 2016, it contributed 3 to 5% of the country's total marine production [8]. It is therefore a species in high demand for human consumption.

Due to the open access fisheries development policy, there was a considerable expansion of fishing capacity from the mid-1970s onwards, which led to overfishing of nearshore fishing grounds in the early 1980s and triggered conflicts between large and small-scale fisheries [9–13]. Between 1990 and 2012, marine fisheries resources off the west and east coasts of Peninsular Malaysia were reported to have been exploited beyond their maximum sustainable levels [14–18]. Indicators of overfishing include increasing difficulty in catching certain fish species, a decline in average catch volume, more time at sea, and a decline in fish composition [15,19,20]. Given the importance of fisheries IM to the region's

food supply and increasing demand [7], it is important to develop effective management measures to ensure the sustainability of stocks to meet human demand and the balance of the marine ecosystems in which they occur. Such management involves the introduction of new strategies or control measures.

The main fishing methods used to catch Indian mackerel in Peninsular Malaysia are purse seiners (51%) and trawlers (28%), as measured by the total catch of Indian mackerel reported in 2016, while the remainder is accounted for by other types of fishing vessels, including artisanal fisheries [8]. In 2016, there were 720 purse seiners and 3900 trawlers in Peninsular Malaysia [8].

To get a better idea of how sustainable fisheries IM can be achieved, this paper aims to assess the optimal conditions required to achieve it. So far, stock assessment of IM has been based on limited data. Maximum sustainable yield (MSY) was determined from landings trends in catch and effort data, while stock assessments were based solely on sampling surveys (using a tagging system). The lack of biological data was one of the limiting factors preventing the determination of IM stock status [1,18,21], and these previous studies have focused only on biological factors and have failed to take a holistic approach that incorporates other factors that may influence fisheries.

To overcome this traditional way of understanding and managing fisheries IM, we have explored a more holistic approach to modelling opportunities for the internal task of managing an open access fishery. To this end, a system dynamics model [22] was used to provide a general framework that incorporates several interacting factors that influence the management of the fishery. In addition, system dynamics is used both quantitatively with simulation models and qualitatively with causal cycle or stock and flow diagrams (further explanation in the stock and flow section), so it can be used for both testing and theory building [23].

Previous applications of systems dynamics modelling in the context of fisheries have addressed competition between fishers, shrimp commodity cycles, strategic planning of fishing groups, management of specific fisheries, understanding resource management concepts, simple and complex models [24], illustrating the value of systems dynamics modelling [25] and a variety of related topics [26]. Recent work by Sigríður et al. [27], Fatma [28], and Brennan et al. [29] has focused on sustainable fish populations. These studies help to broaden the basis for holistic fisheries management, which is essential for protecting fish stocks to maintain food security and ecological integrity.

System dynamics is proving to be a useful tool for understanding the complex interactions between social, political, economic, ecological, and biological factors in a fisheries system over time. Therefore, it is practical and applicable to use such a model in the case of fisheries IM in Peninsular Malaysia. In this study, an integrated model is constructed to simulate the functioning of the IM fishery under various open access conditions. The model consists of three sub-models, namely (1) the biological, (2) the economic, and (3) the management model of the system. Such a model surpasses traditional methods of dealing with fisheries IM [22] by incorporating a wider range of factors considered. It is hoped that this approach will stimulate further research and investigation into fisheries management systems in the region.

By incorporating historical and current data on the population of IM, catch per unit effort (CPUE), price, and other important factors at different fishing capacities, the model aims to provide valuable insights into the status of stocks when different management measures are taken to limit fishing capacity. This is performed by limiting the number of boats allowed or controlling the fishing gear used. By improving information and examining different management options, this article offers recommendations for management measures that will increase the overall efficiency of the IM fishery.

Section 2 presents the development of the system dynamics approach to modelling IM in Malaysia and reviews all processes and inputs used in the development of the model. Section 3 explains the process of validating the model based on the work of Forrester and

Senge [30]. The results and discussion of the model are detailed in Section 4 and the policy implications and conclusions are presented in the final section of this paper.

## 2. Method

The structure of the system dynamics model is determined by the concept from the written literature, the purpose, and the mental and written information. The structure is connected to form a model according to the principle of feedback loops. After the model is created, the behaviour from the model is compared to the behaviour in the real world (based on historical data). If there are discrepancies between the model and the real world, the model needs to be improved in terms of its structure and parameters. Then, it can be used for policy evaluation and also for policy alternatives. There are five steps to look at a problem from a system dynamics perspective [17,31,32]. The steps are as follows: (1) problem articulation; (2) formulation of the dynamic hypothesis; (3) formulation of a simulation model; (4) testing; and (5) policy design and evaluation. The dynamic system IM was developed using Vensim software [17,33].

Vensim provides a simple and flexible environment for building simulation model from the causal loop diagram, as well as presenting it using stock and flow diagram. By connecting variables with arrows, relationships among system variables are entered and recorded as causal connections. The model is analysed throughout the building process, looking at the causes and uses of a variable, and looking at the loops involving a variable [33]. After completion of the model development, the model is simulated, and the behaviour of the model is deliberated [33]. The advantage of using Vensim software is it is easy for the modeller to build the model since the software provides icons. However, the drawback of using vensim is it is more complicated compared to other software, i.e., Stella, iThink because it has more features and for that reason it takes more time to get used to.

### 2.1. Problem Articulation: IM Model Boundary

The system dynamics model for IM in Peninsular Malaysia (Figure 1) consists of interconnected modules that describe biological, economic, and management aspects. Table 1 provides a list of key variables that have captured the dynamic behaviour of IM in Peninsular Malaysia. There are two types of variables, endogenous and exogenous. Endogenous variables change in their behaviour patterns when the model structure or decision rules are changed. The exogenous variables are determined outside the model structure. There are no feedbacks to exogenous variables. However, some exogenous variables have a significant influence on the endogenous variables. The time horizon ranges from 1980 to 2050 and covers 36 years of historical data and 38 years of future projections. This time window provides sufficient time to assess the short- and long-term impacts of policy changes on the key variables.

**Table 1.** Model Key Variables.

| Endogenous | Exogenous |
| --- | --- |
| IM Juvenile | Batch per IM adult |
| IM Adult | Reference IM Adult |
| Fishing mortality (catch) | Boat discard rate |
| CPUE | Normal CPUE (Normal CPUE fraction in the IM model = 12. This is calculated based on average catch of IM per boat (Annual Fisheries Statistics Malaysia, 1980–2016). |
| Total number of boats | Cost per boat |
| Total operation cost | Average boat lifetime |
| Expected profitability | |
| Revenue | |
| Price | |

(i)    Sample data collection

The endogenous and exogenous variables were obtained from secondary data or estimated based on historical data. In this study, the historical data of the key variables are fishing mortality (catch), CPUE, total number of boats (trawlers and purse seiners), and price. These data are from the annual fisheries statistics of Malaysia.

(ii)    Model assumptions

The model contains several assumptions to facilitate the modelling process. It is assumed that these assumptions do not compromise or undermine the adequacy of the model or the robustness of the model structure. The main reason for using these assumptions is to focus on the dynamic behaviour of the key variables.

The assumptions of the model are as follows:

(1)    the effect of climate change on the density of IM is assumed to be 5% in the model, which will affect fish reproduction.
(2)    the biomass of adult fish from IM is initially estimated at 60,000 tonnes in 1980. This is based on the behaviour of fishing mortality (catch), total number of boats and CPUE.
(3)    Juvenile biomass is initially estimated at 30,000 tonnes, based on the adult biomass estimate from IM.
(4)    The cost per boat (labour, insurance, license, fuel, etc.) was taken as a constant as no data was available and is based on the study "The Impact of Artificial Reefs on Fishers' Catches in Malaysia" [34].
(5)    The growth share of the price was estimated based on aggregated price data from Malaysia's annual fisheries statistics.
(6)    Gear efficiency is considered constant and assumed to be 5%.

*2.2. Formulation of Dynamic Hypothesis*

The dynamic hypothesis of system dynamics in this model is established by postulating the causes of the problem in a theory that is tested with simulations. The word "dy-namic" indicates that the problem results from a feedback structure.

**Hypothesis 1.** *Limiting the number of fishing boats, which also means reducing the number of days fishermen fish, will increase catches and CPUE.*

**Hypothesis 2.** *The increase in climate change will affect fish density, which will increase the breeding season, which in turn will affect the breeding rate (lower). Therefore, the total number of IM juvenile fish will be lower, reducing the population of IM.*

*2.3. Formulation of a Simulation Model*

2.3.1. Causal Loop Diagram of the IM Model

Causal loop diagrams (CLDs) are an important tool for showing the feedback structure of systems. CLDs are best suited for capturing hypotheses about the causes of dy-namics, provoking and capturing the mental models of individuals or teams, and linking the important feedbacks that contribute to a problem. A CLD contains variables that are connected by causal links represented by arrows. These links denote the causal effects between the variables. In this example, the IM birth rate is determined by both the population and the IM birth rate. Each causal link is given a polarity, either positive (+) or negative (−), to indicate how the dependent variable changes when the independent variables change. The important loops are highlighted with a loop identifier to indicate whether the loop has positive (reinforcing-R) or negative (balancing-B) feedback. Note that the loop identifier (R or B) circulates in the same direction as the loop it corresponds to.

(a)    Biological sub-model

The first reinforcement loop is triggered by the growth rate of the young (Figure 1). When the maturation rate increases, the number of IM adults (stock) also increases (R1).

Therefore, reproduction among the IM adults also increased. The second reinforcement loop (R2) works through fish density. Higher IM density values increased spawning time, which affected the spawning rate among the IM. This increased the number of IM juveniles and the stock. However, when both fishing mortality (catch) (B1) and natural mortality increased (B2), the number of IM juveniles decreased (R3).

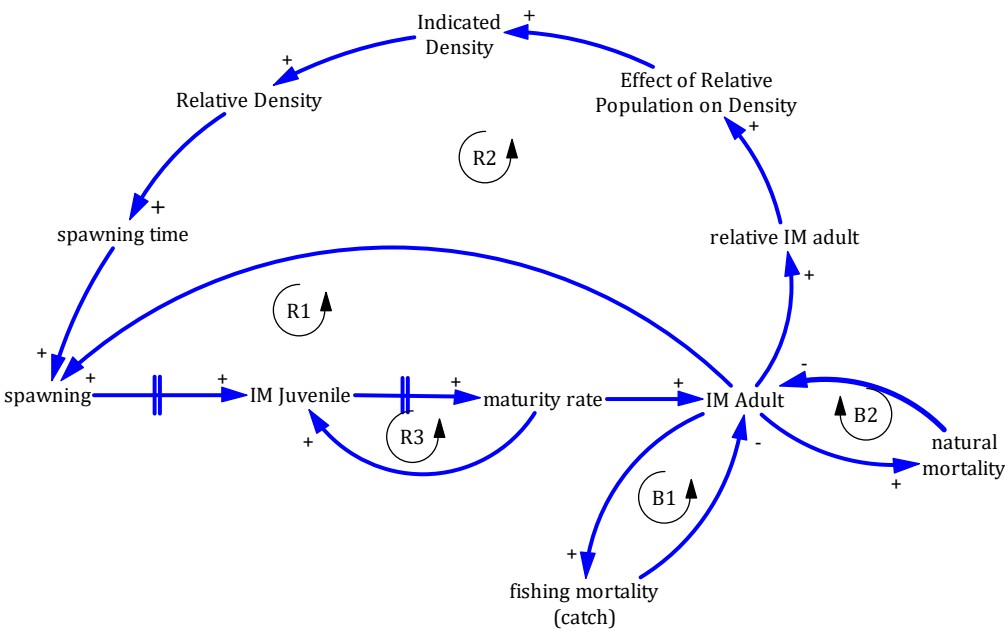

**Figure 1.** Causal Loop Diagram for IM Population Growth.

(b)    Management sub-model

Figure 2 shows the variables that affect the total number of boats where when the discard rate increased the number of boats is decreased (B4). When the total number of boats increased, both catch and fishing mortality (catch) increased (R4), which improved CPUE and resulted in more boats going out to fish. Despite the enhancement, CPUE declined when fishing capacity (boats) increased (B3).

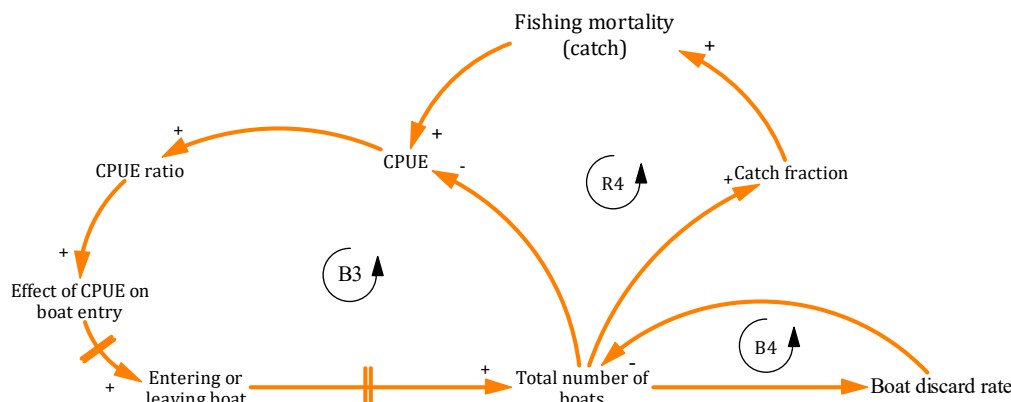

**Figure 2.** Causal Loop Diagram for the Total Number of Boats.

(c)    Economic sub-model

Figure 3 shows the causal cycle for the price of IM. As the price increased, so did the price change (R5). Therefore, both revenue and expected profitability increased.

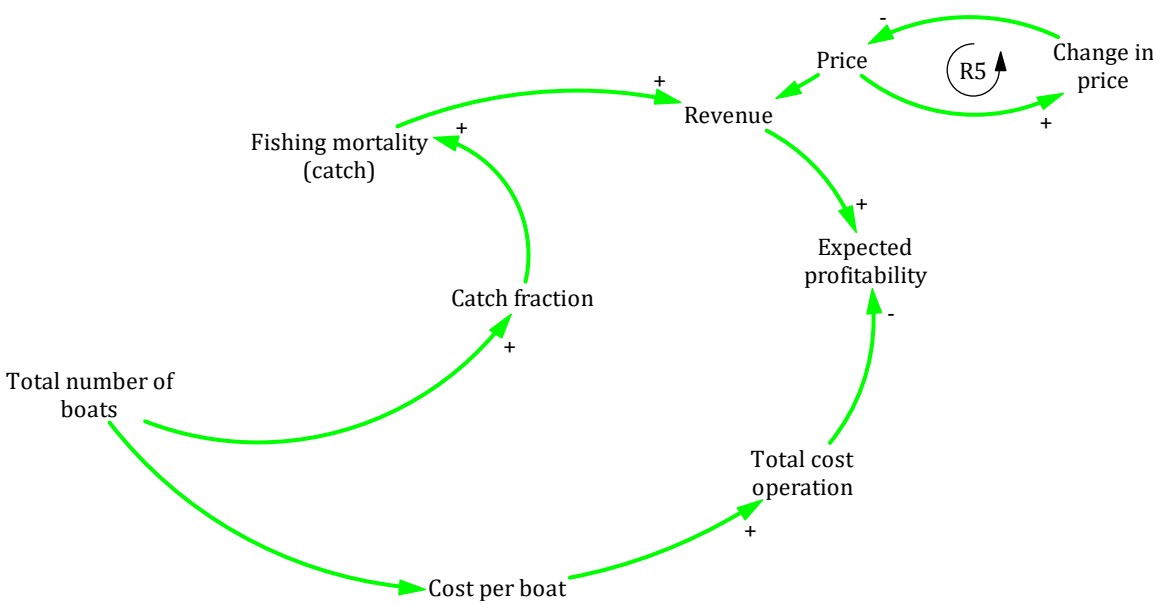

**Figure 3.** Reinforcing and Balancing Loop for Price.

2.3.2. Stock and Flow Diagram of IM

The IM stock and flow diagram defines the underlying physical structure and decision rules in mathematical equations. Stocks describe the state of the system and provide information about it. Flows provide adjustments to the stocks through addition and subtraction over time.

The best way to build a system dynamics model is to start with the stocks, add the flows and then use converters to explain the flows. Stocks in system dynamics models are accumulations [24,35]. They characterise the state of the system and generate the information on which decisions and actions are based [35]. The conventions of stock and flow diagrams originate from Forrester in 1961 and are based on a hydraulic metaphor—the flow of water into and out of reservoirs. The diagram notation for stocks and flows is referred to as below: Stocks are represented by rectangles (visualizing a container holding the contents of the stock).

- Inflows are represented by a pipe with arrows pointing into the supply (addition) and outflows are represented by pipes pointing out of the supply (withdrawal).
- Valves are used to control the flows.
- The clouds represent the sources and sinks for the flows. A source denotes the stock from which a flow originating outside the model boundary originates; sinks denote the stocks into which flows leaving the model boundary drain. Both sources and sinks are assumed to have unlimited capacity and can never restrict the flows they support.

For an example of a stock and flow diagram, see Figure 4. As the example in Figure 4 shows, an IM stock is a stock (IM adult) that accumulates the inflow of maturation rate and is reduced by the outflow of mortality rate. These are the only flows considered in the model (unless explicitly stated, other possible flows into or out of the stock are assumed to be zero). The clouds show that the stock of IM adults never suffers from the maturation rate and the stock of IM adults never becomes so high that it blocks the mortality rate.

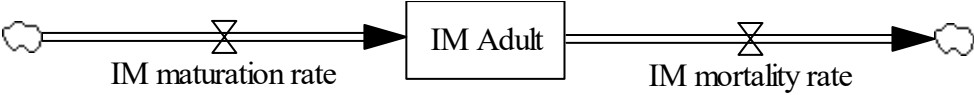

**Figure 4.** Stock and Flow Diagramming Notation.

The stock and flow diagram has a precise and unambiguous mathematical meaning. Stocks accumulate or integrate their flows; the net flow into the stock is the rate of change of the stock. Therefore, the structure shown in Figure 5 corresponds exactly to the following integral equation:

$$\text{Stock (t)} = \int_{t_0}^{t} [\text{Inflow(s)} - \text{Outflow(s)}]ds + \text{Stock (t}_0)$$

where Inflow(s) indicates the value of the inflow at any time s between the initial time $t_0$ and the current time t. Equivalently, the net rate of change of any stock, its derivative, is the inflow less the outflow, defining the differential equation:

$$\frac{d(\text{Stock})}{dt} = \text{Inflow(t)} - \text{Outflow(t)}$$

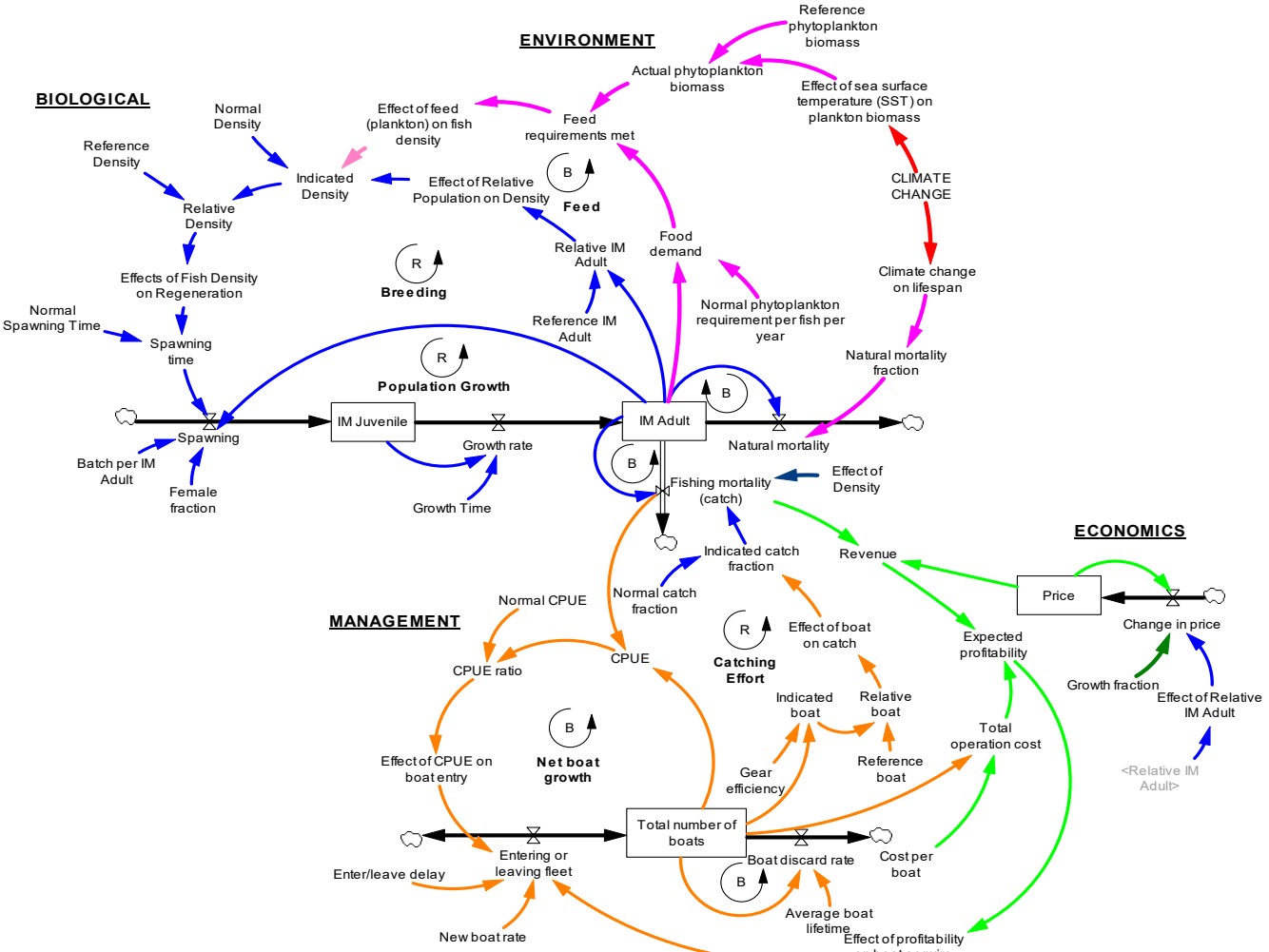

**Figure 5.** Stock and Flow Diagram of IM.

In general, the flows will be functions of the stock and other state variables and parameters [17].

As stated in Section 2.3.2, the stock and flow diagram define the underlying physical structure and decision rules in mathematical equations (Appendix A). Stocks are accumulations of IM adult, IM juvenile, and total number of boats. Stocks describe the state of the system and provide information about it. Flows make adjustment in stocks through addition and subtraction over time. Flows include catch, breeding, natural mortality, entering

or leaving boats, and boat discard rate. This is shown in Figure 5. The IM Juvenile stock is dependent on the breeding rate and the growth rate. It is assumed that the initial value for IM Juvenile stock is 3000 tonnes. The formulation of the stock is given below:

$$\text{IM Juvenile} = \text{INTEG (Spawning} - \text{Growth rate, 30,000)}$$

The growth rate determines the stock of IM adult with an assumption of 60,000 tonnes of stock in 1980.

$$\text{IM Adult} = \text{INTEG (Growth rate} - \text{"Fishing mortality (catch)"} - \text{Natural mortality, 60,000)}$$

The total number of boats stocks is the inflow of entering or leaving boat and an outflow of boat discard rate. The initial value of total number of boats in 1980 is 5000 boats.

$$\text{Total number of boats} = \text{INTEG (Entering or leaving fleet} - \text{Boat discard rate, 5323)}$$

The price of IM is determined by the change of price, for which the rate of change is calculated based on the time series of price data from 1980 to 2016. The initial value for price is USD 213 per tonne.

### 2.4. Testing (Model Validation), Policy Design, and Evaluation

Forrester and Senge [30] presented a number of different informal and formal tests and discussed that confidence in a model can be built to the extent that it passes these tests. The author conducted a series of experiments that are commonly used and considered most important: Tests to verify structures/parameters, extreme conditions, and behavioural tests. All these tests were carried out in Vensim software. Figures 6 and 7 show the comparison between the predicted and historical behaviour of fishing mortality and total number of boats in Malaysia. The modelled simulations of fishing mortality and total number of catches agree reasonably well with the historical behaviour, and the model is shown to be reliable. The validated model has been used for baseline scenarios and policy analysis. Figure 8 shows Theil's inequality is a quantitative check of how the simulated and the actual data fit can be done. The inequality test is explained using diagrams to define the errors through the decomposition technique for each variable. This provides an easily interpreted breakdown of the sources of error by dividing the mean square error (MSE) into three components: bias ($U^M$), unequal variation ($U^S$), and unequal co-variation ($U^C$) [17].

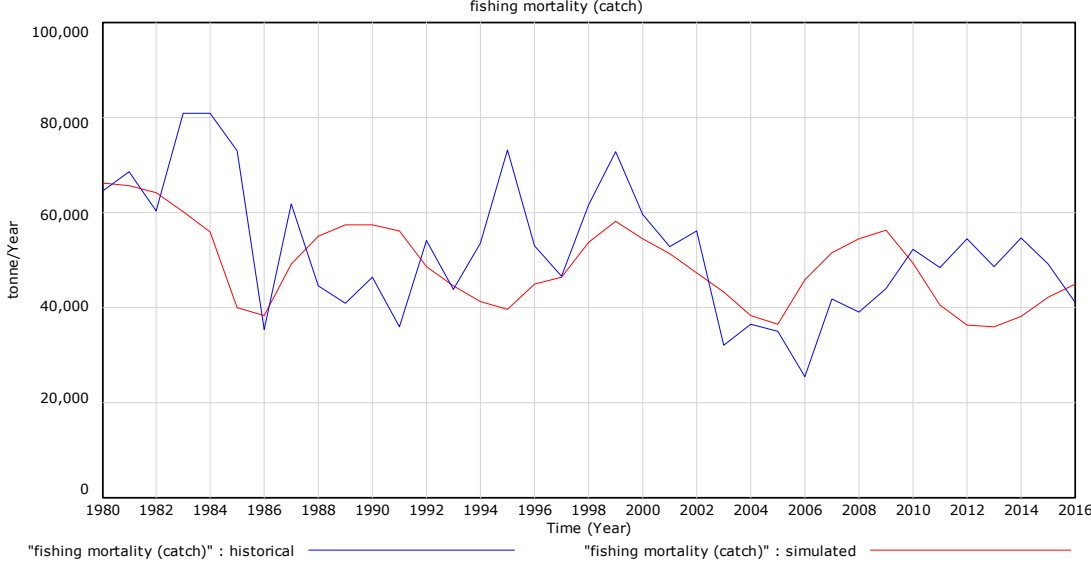

**Figure 6.** Simulated and historical data of fishing mortality (catch) in Malaysia.

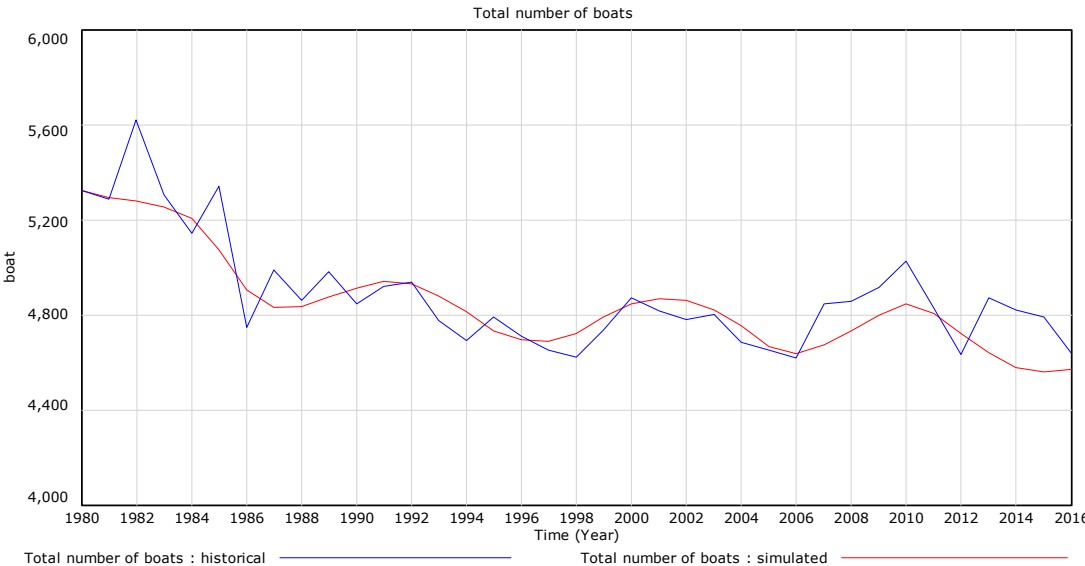

**Figure 7.** Simulated and historical data of total number of boats.

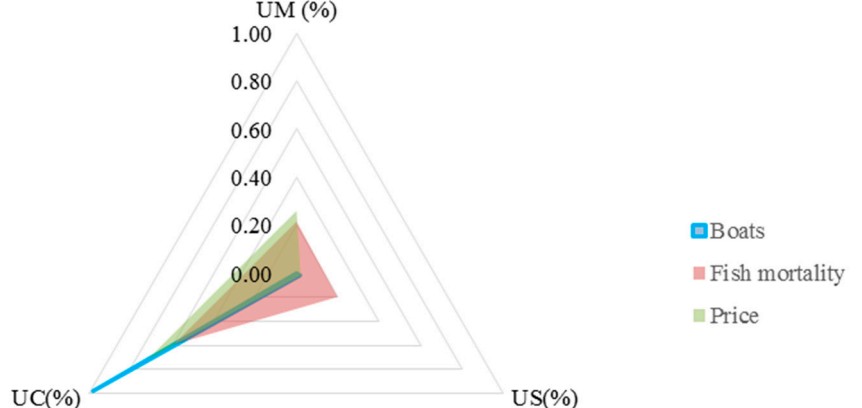

**Figure 8.** Theil Inequality Statistics.

The Theil statistic shows the total error and how the error is decomposed into the components of bias (UM), unequal variation (US), and unequal covariation (UC) [17]. The error in fish mortality due to bias and unequal variation was 22% and 20%, respectively. This means that the simulated and historical behaviour have similar mean values and show a similar pattern. Most of the error (59%) is in the unequal covariation and results from the pointwise variation around the simulated result (Figure 6).

The error in the number of boats due to bias, unequal variation, and unequal covariation was 0%, 2%, and 98%, respectively. The error due to unequal variation is zero. Most of the error is due to unequal covariation. This means that the model captures well the mean and the main changes in the actual data series. The general behaviour of the simulated result agrees well with that of the actual data (Figure 7).

The source of information for the parameters in system dynamics models can be any accessible data source: physical laws, accurate experiments, empirical data, expert decisions, and even individual intuition [36,37]. Therefore, system dynamics models inherently contain a large set of highly uncertain parameters as the model increases in scale and depth. Assessing trend sensitivity, also known as statistical screening of system dynamics models, assists in finding the most important of the uncertain parameters that perturb the behaviour of key variables. The most important parameters can be identified by estimating the correlation coefficients between the simulated output and the values given to each input in a sensitivity test using Vensim software [36,37].

In accordance with the Vensim User's Guide, sensitivity testing is a procedure in which model expectations are changed with respect to the value of constants in the model and the final result is tested. A parameter is an alternative expression for a constant. This way, modellers can identify the most important parameters of the model. This identification is of great value in accepting the impact of each parameter used in the model and in developing different strategies to adjust the assumptions for the parameter values. It can also examine the efficiency of alternative strategies after simulation runs at the strategy proposal stage. The parameters that can affect the stock of IM are listed in Table 2. The parameters are given with their current model values and the minimum and maximum values to be tested.

**Table 2.** List of Parameters for the Sensitivity Analysis of the IM stock.

| No. | Parameter | Model Value | Min | Max |
|:---:|:---:|:---:|:---:|:---:|
| 1. | Average boat lifetime | 8 | 7 | 9 |
| 2. | New boat rate | 7500 | 7300 | 7600 |
| 3. | Enter/leave delay | 1 | 0.6 | 5 |
| 4. | Reference IM Adult | 60,000 | 58,000 | 80,000 |
| 5. | Cost per boat | 2272 | 2000 | 2300 |
| 6. | Growth Time | 0.5 | 0.45 | 0.55 |

A total of 6 parameters were run 50 times with a random uniform distribution of input values between the minimum and maximum over 33 periods using Vensim software. The results of the sensitivity analysis are shown in Figures 9 and 10. The 100% percentile is shaded to account for all 50 simulation runs. The 95% percentile is obtained by removing the lowest 2.5% and the highest 2.5% of the simulation runs. The 50% percentile is determined by taking out the lowest 25% and the highest 25% of the runs.

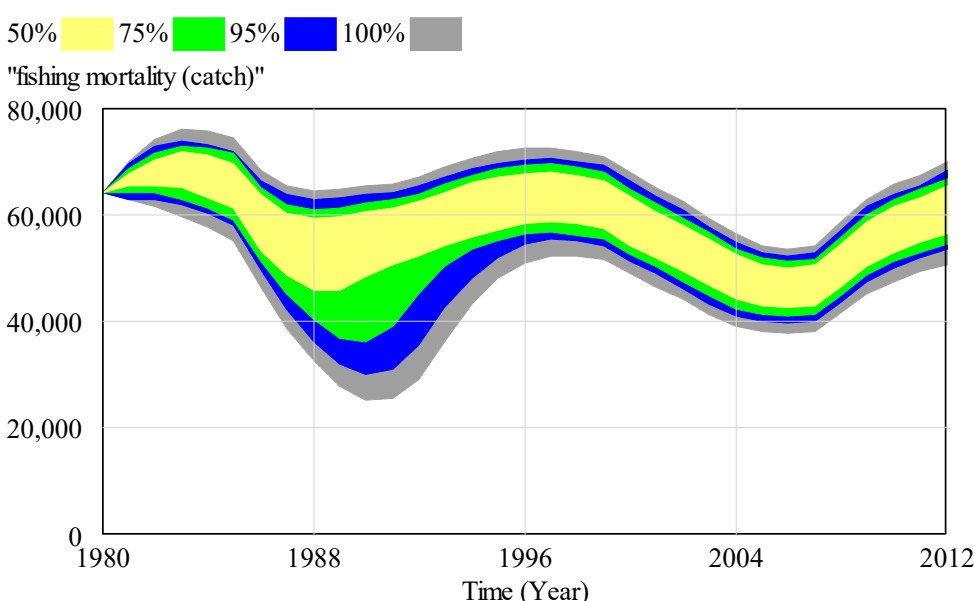

**Figure 9.** Sensitivity of fishing mortality (catch).

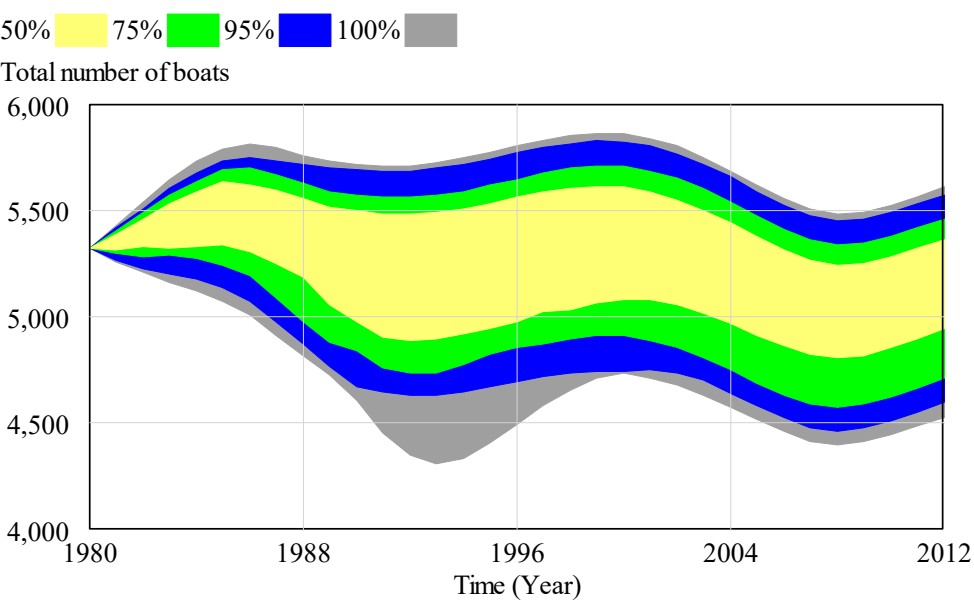

**Figure 10.** Sensitivity of total number of boats.

## 3. Results

Fisheries systems involve complex interactions between resource stocks and the people involved in exploiting those stocks. Therefore, the dynamics of fish stocks are an important factor in determining appropriate fisheries management policies. Catch per unit effort (CPUE) remains one of the most important concepts in fisheries management and research and is considered the main factor influencing stock dynamics and stock productivity. Therefore, the objective of this study is to recommend a better management strategy to conserve the stock of IM. This section focuses on the policy analysis based on the open access scenario with changes in the key variables of total number of boats.

### 3.1. Simulation Analysis for an Open-Access Scenario

'Open access', also known as 'business as usual' (BAU), is the condition in which access to fishing (to harvest fish) is unrestricted, i.e., the right to catch fish is free and open to all. The simulation of the situation of free access to fisheries IM in Peninsular Malaysia was conducted without any changes or policy restrictions. It was conducted from 1980 to 2050, and all baseline values used in the simulation were the same values and situations as in 2016. The results of the model simulation under the open access scenario for IM are shown in Figure 11. They show that the stock of IM decreases slightly from 2015 to 2050, with minimum and maximum values of 58,000 tonnes and 60,400 tonnes, respectively. The catch of IM also decreases, which is associated with a decrease in the number of boats from 2016 to 2050. This suggests that catches will remain the same in the future, even if there are fewer boats. This is likely due to advances in modern technology, such as the use of GPS by fishermen. The result of a decrease in the number of boats suggests that people may have less interest in fishing in the future due to low incomes and higher boat maintenance costs. With fewer boats, the simulation also shows an increase in annual CPUE for IM. Despite the lower catch, the increasing trend of IM price indicates that IM will continue to be in high demand among Malaysians.

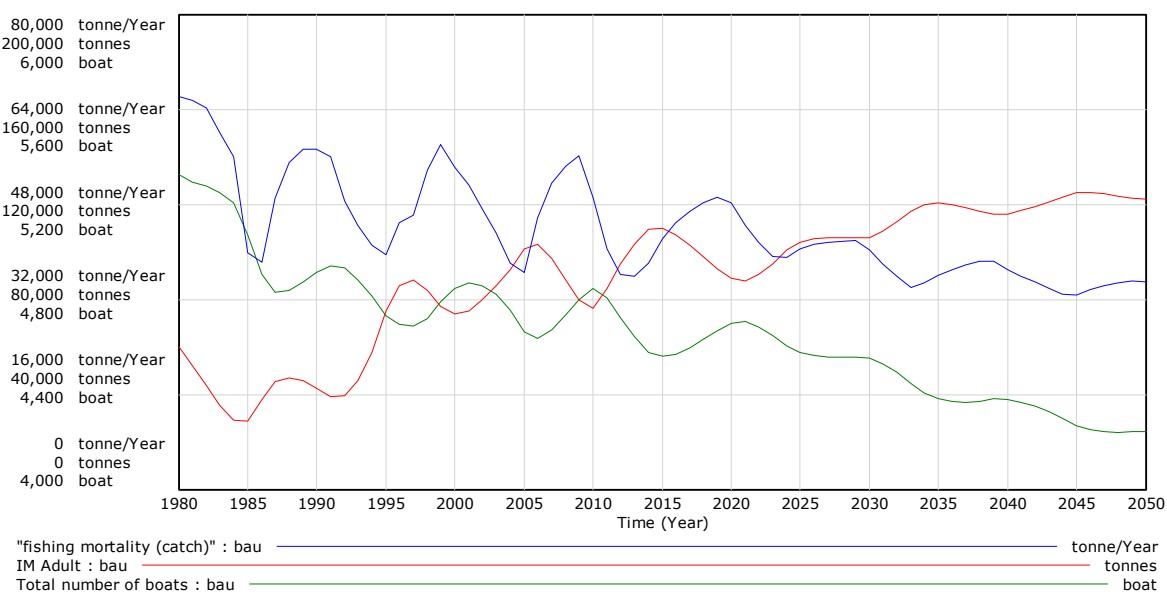

**Figure 11.** Open-access simulation results.

### 3.2. Simulation Analysis for Alternative Management Policy Scenarios

Based on the results of the free access simulation of IM, an alternative policy is implemented. The impact of the policies is evaluated, and the appropriate combination of policies are selected as a recommendation for fisheries management. The alternative policy aims to reduce the number of fishing boats, which directly leads to a reduction in the number of fishing days. The policy also considers the overall performance of the boat ("gear efficiency"), making assumptions about the rate of reduction. The details of the alternative policy scenarios are discussed in this chapter. A summary of the alternative policy scenarios is provided in Table 3.

**Table 3.** Policies for Simulation.

| Policy Scenario | Number of Boats | Gear Efficiency |
|:---:|:---:|:---:|
| A | 25% reduction | 10% reduction |
| B | 50% reduction | 15% reduction |
| C | 75% reduction | 25% reduction |

### 3.3. Policy Scenario A

Limiting the number of boats by 25% and gear efficiency by 10% shows that the stock of IM is at sustainable levels in 2050 compared to the business-as-usual (BAU) scenario. Catches were slightly higher after 2016, which also increased CPUE (Figure 12). The increase in fish stock eventually led to a decrease in fish price, which decreased exponentially by 3% compared to the BAU scenario.

### 3.4. Policy Scenario B

Limiting the number of boats by 50% and gear efficiency by 15% shows that the stock of IM is higher compared to the Policy A scenario. With this reduction in the number of boats, the stock gradually increased while the catch and price decreased by 7% and 6%, respectively, compared to the BAU scenario. CPUE was almost 40% higher in this scenario than in the BAU scenario (Figure 13).

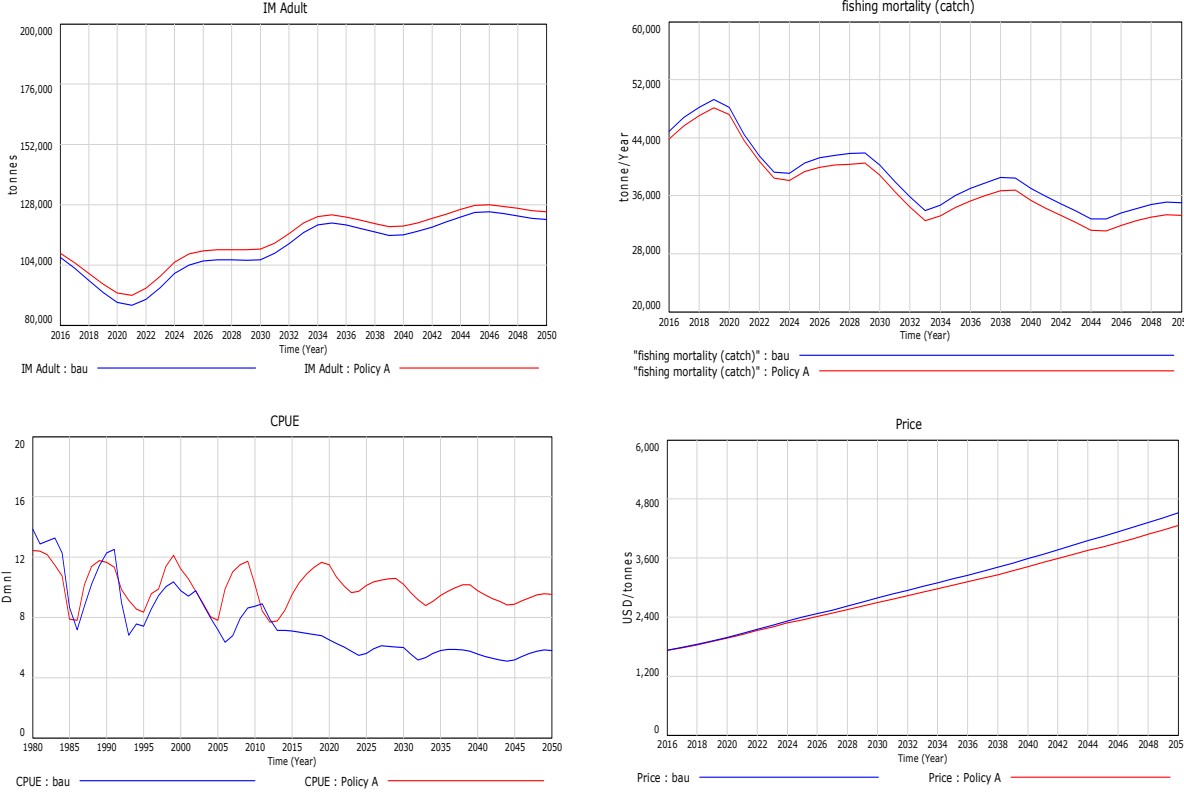

**Figure 12.** Policy scenario A.

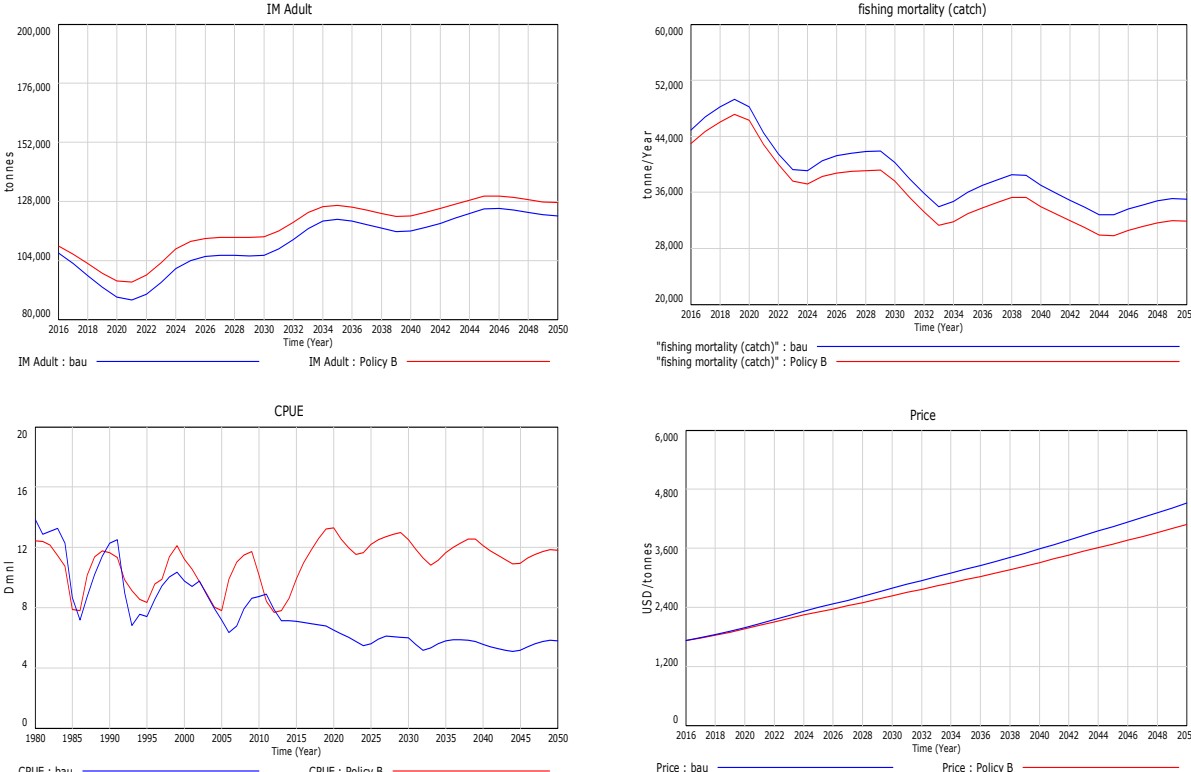

**Figure 13.** Policy scenario B.

### 3.5. Policy Scenario C

With a 75% reduction in the number of vessels (trawler phase-out) and a 25% reduction in gear efficiency, there is little change in fish stock and catch (Figure 14). However, CPUE is slightly higher when only purse seiners are fishing. Overall, the results indicate that a drastic reduction in the number of fishing vessels or fishing days has a significant impact on the fish stock of IM or the improvement of catches.

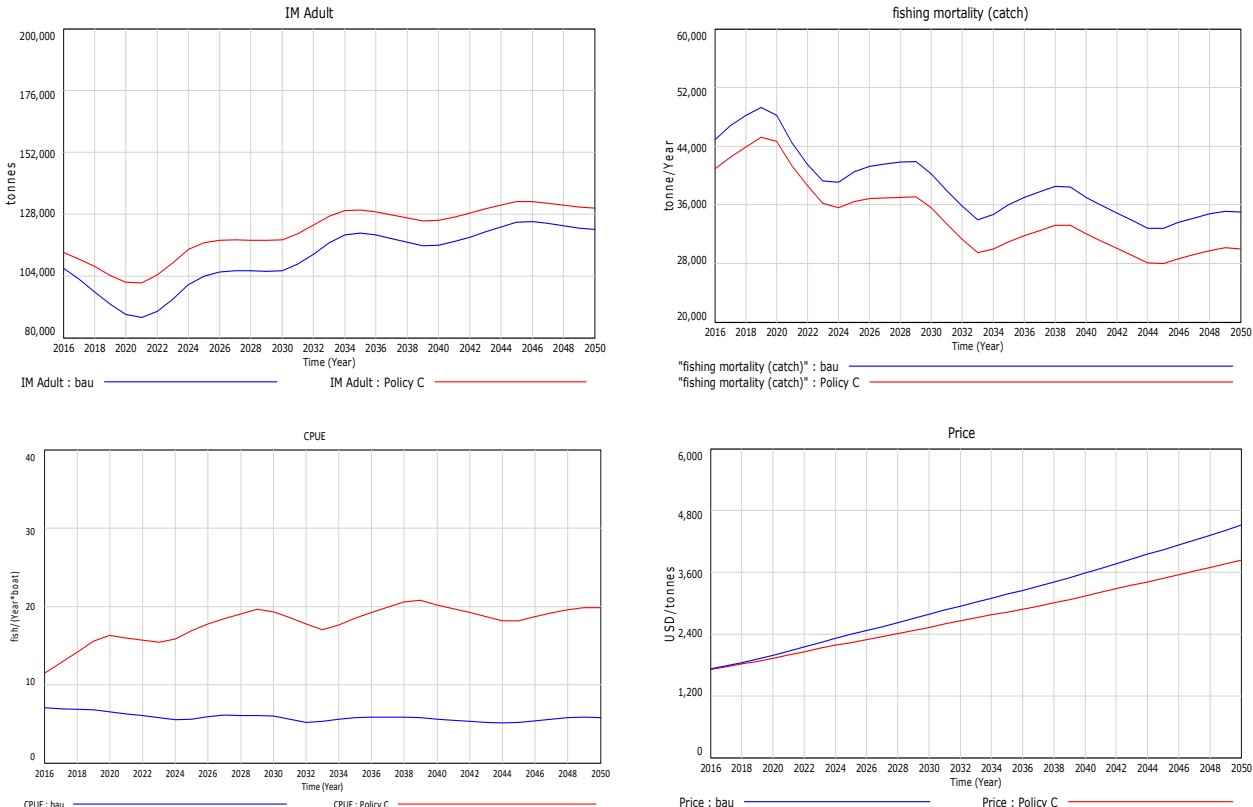

**Figure 14.** Policy scenario C.

Overall, of the measures proposed in the simulation model, the drastic reduction in the number of boats (measure C) has a greater impact or change on the stock than the smaller reductions in the number of boats (as in measure scenarios A and B). This could also be due to the fact that the low number of catches allows the adult IM more time to spawn (at least 3–4 times per year, producing about 650,000 eggs per spawning event). The impact of the lower number of boats will also lead to a positive increase in CPUE to 10–20 tonnes (in all three scenarios), compared to 6–7 tonnes in the BAU simulations. This will therefore contribute to more landings and have an impact on the price of IM (as shown in Figure 15).

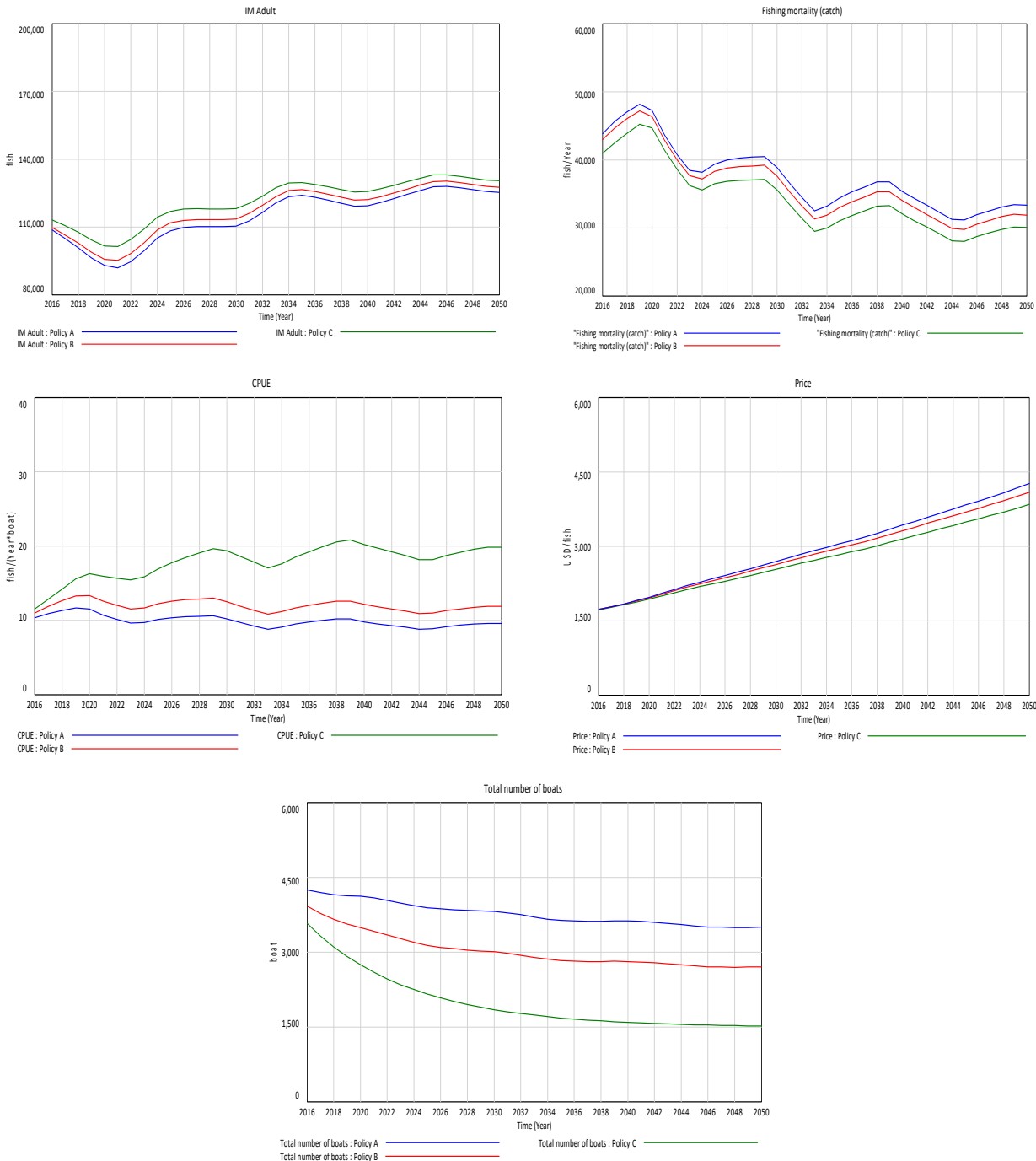

**Figure 15.** Behaviour of IM Adult, Fishing Mortality, CPUE, Total Number of Boats, and Price for Policy scenarios A, B, and C.

## 4. Discussion

The results of applying the first scenario (A) with a 25% and 10% reduction in the number of boats and gear efficiency, respectively, show that the stock of IM maintains a sustainable level compared to the business-as-usual scenario, suggesting that some reduction in fishing capacity is required to ensure the longevity of this halieutic resource. However, the second and third policy scenarios (B and C) show that the stock IM and catch per unit effort are higher with a large reduction in fishing capacity, with CPUE exceeding the normal level for the catch IM. Although Scenario A shows a sustainable level, Scenarios B and C are simulated to show the worst-case scenarios in terms of fishing

capacity limitation for the IM system (to show the impact on other key variables, i.e., fishing mortality, CPUE and number of boats). These scenarios serve as important information for policy makers to determine how many fishing boats should be taken out of service to maintain the stock of IM and thus determine the amount of catch to meet the demand of IM. The summary of the simulation of all scenarios is shown in Table 4.

**Table 4.** Summary of the simulations.

| Time (Year) | IM Juvenile Stock (Tonne) | | | |
| --- | --- | --- | --- | --- |
| | BAU | Policy A | Policy B | Policy C |
| 1980 | 30,000 | 30,000 | 30,000 | 30,000 |
| 2016 | 7735 | 21,272 | 21,105 | 20,571 |
| 2050 | 13,870 | 17,715 | 17,048 | 16,160 |
| Time (Year) | IM Adult Stock (Tonne) | | | |
| | BAU | Policy A | Policy B | Policy C |
| 1980 | 60,000 | 60,000 | 60,000 | 60,000 |
| 2016 | 112,384 | 108,566 | 109,719 | 113,003 |
| 2050 | 154,608 | 125,101 | 127,432 | 130,361 |
| Time (Year) | Total Number of Boats (Boat) | | | |
| | BAU | Policy A | Policy B | Policy C |
| 1980 | 5323 | 5323 | 5323 | 5323 |
| 2016 | 4616 | 4247 | 3918 | 3567 |
| 2050 | 4351 | 3494 | 2698 | 1515 |
| Time (Year) | Fishing Mortality (Catch) (Tonne) | | | |
| | BAU | Policy A | Policy B | Policy C |
| 1980 | 73,751 | 66,165 | 66,165 | 66,165 |
| 2016 | 32,454 | 43,800 | 43,000 | 40,946 |
| 2050 | 25,240 | 33,287 | 31,884 | 30,016 |
| Time (Year) | CPUE (Fraction) | | | |
| | BAU | Policy A | Policy B | Policy C |
| 1980 | 14 | 12 | 12 | 2 |
| 2016 | 7 | 10 | 11 | 11 |
| 2050 | 6 | 10 | 12 | 20 |
| Time (Year) | Price (USD) | | | |
| | BAU | Policy A | Policy B | Policy C |
| 1980 | 213 | 213 | 213 | 213 |
| 2016 | 1896 | 1725 | 1724 | 1720 |
| 2050 | 4063 | 4261 | 4089 | 3840 |

From the findings, we suggest the following overall policy implications:

- The relationship between gear efficiency and boat performance is directly related to the number of fish caught. Therefore, when boats reach their full capacity (i.e., new boat, net mesh size), they catch IM to their maximum capacity. When trawlers in Peninsular Malaysia exit the fishery (Policy C), CPUE is 20% higher for purse seiners and there are positive changes in fish stock and catches. Therefore, the government should consider strictly issuing boat licences, especially for trawl boats, by demarcating the area and monitoring the number of boats allowed to operate.

- The increased CPUE affects the price of IM. So, there is a significant relationship between the catch and the price. From this study, the larger the catch of IM, the lower the price of IM, which is beneficial to Malaysians as IM is a popular fish. Since IM in Peninsular Malaysia is related to phytoplankton content, it is important for the government to ensure that the coastal health quality is in the best condition by taking measures to prevent erosion, philtre pollutants, and provide food, shelter, breeding grounds, and nursery areas for a variety of organisms.

The contribution of this paper is twofold, both theoretical and managerial. The general contribution of the research is to introduce modelling methods from the field of engineering into the fisheries context. In addition, the research provides a novel approach to time series simulation that differs from typical statistical methods in that it does not require the statistical assumptions necessary to fit the data to parametric time series modelling.

In terms of modelling method, simulation models have been used in the past to support fisheries management. This paper presents a new modelling method for simulating Malaysian fisheries data, a method that is not widely used in the Malaysian fisheries modelling field, and furthermore, this model can be easily updated and adapted to current data series or changes in policy and management. Even though a system dynamics model is a comprehensive method for analysing the system IM, a practitioner should be aware of one particular drawback: The simulation in the model cannot provide accurate results if most of the input data is based on assumptions. However, the model is sufficient to represent the fundamentals of the IM system under different measures. This IM system dynamics model is also applicable to other fish species for which the status of the stock is not known.

It should be noted that the above policy analysis is only valid within the framework of the model developed in this paper. The limitations of the model therefore also restrict the scope of the analysis. The following limitations, as well as the key assumptions of the model, which are shown in Figure 5, should be taken into account when evaluating this policy analysis: (1) The model used for the policy analysis has been largely validated qualitatively and quantitatively in terms of its feedback structure, internal bias, and response to exogenous policies. Therefore, while it is argued that the model is a valid representation of the Indian mackerel system in Peninsular Malaysia, it is not argued that the model is a tool for quantitative prediction. (2) The Indian mackerel system dynamics model has selective limitations that depend on the objective of the model. The objective is to provide a baseline model that can endogenously produce realistic behaviour in response to policy and structural changes. However, there is one key effect that has not been well integrated and extended into the model, namely the impact on ecosystem health. This key variable is important in determining the lifespan and the density of Indian mackerel. Here in the model, the climate change impact variables only acted as a switch by assuming a rate of 5%. (3) The number of boats can be expanded to include gear type and catch by area, which will have a greater impact on the stock.

To model the reality of system behaviour more comprehensively, this model should be further developed with a sub-model for ecosystem health impacts associated with varying fishing pressure and climatic changes. This would improve the reliability and robustness of the model structure.

**Author Contributions:** Conceptualization, I.I.; Data curation, I.I.; Formal analysis, I.I.; Methodology, I.I.; Resources, P.F.; Supervision, P.F. and A.T.; Writing—original draft, I.I.; Writing—review & editing, I.I., P.F., A.M. and A.T. All authors have read and agreed to the published version of the manuscript.

**Funding:** This research was funded by Ministry of Higher Education Malaysia (MOHE), Universiti Putra Malaysia (UPM), and University of Portsmouth (UoP), for the PhD research scholarship.

**Institutional Review Board Statement:** The study was conducted according to the guidelines of the Research Ethical Review, and approved by the Ethics Committee, of UNIVERSITY OF PORTSMOUTH (protocol code E333 and 2 March 2015).

**Informed Consent Statement:** Not applicable.

**Data Availability Statement:** This study did not report any data.

**Acknowledgments:** We would like to express our sincere thanks to Ministry of Higher Education Malaysia (MOHE), Universiti Putra Malaysia (UPM), and University of Portsmouth (UoP), for the PhD research scholarship provided and for allowing us to embark on this study on modelling the Indian mackerel off Peninsular Malaysia. We hope this study will open up more explorations or expansion of this work to look into more complex issues such as the impact of climate change and environmental issues.

**Conflicts of Interest:** The authors declare no conflict of interest.

## Appendix A

Equations in IM System Dynamics Model:

$$\text{Actual phytoplankton biomass} = \text{Reference phytoplankton biomass} \times \text{"Effect of sea surface temperature (SST) on plankton biomass"} \text{ (Units: ton/Year)} \tag{A1}$$

$$\text{Average boat lifetime} = 8 \text{ (Units: Year)} \tag{A2}$$

$$\text{Batch per IM Adult} = 2 \text{ (Units: fish/fish)} \tag{A3}$$

$$\text{Boat discard rate} = \text{Total number of boats/Average boat lifetime (Units: boat/Year)} \tag{A4}$$

$$\text{Change in price} = \text{Price} \times \text{Growth fraction} \times \text{Effect of Relative IM Adult (Units: USD/(Year} \times \text{fish))} \tag{A5}$$

$$\text{CLIMATE CHANGE} = 1 \text{ (Units: Dmnl)} \tag{A6}$$

$$\text{Climate change on lifespan} = \text{WITH LOOKUP (CLIMATE CHANGE, ([(0,0)-(1,1)], (0,1), (1,0.05))) (Units: 1/Year)} \tag{A7}$$

$$\text{Cost per boat} = 2272 \text{ (Units: USD/boat/Year)} \tag{A8}$$

$$\text{CPUE} = \text{"Fishing mortality (catch)"/Total number of boats (Units: fish/boat/Year)} \tag{A9}$$

$$\text{CPUE ratio} = \text{CPUE/Normal CPUE (Units: 1)} \tag{A10}$$

$$\text{Effect of boat on catch} = \text{WITH LOOKUP (Relative boat, ([(1,0)-(14,0.2)], (1.18349,0.0364912), (4.29969,0.0877193), (13.1835,0.175439))) (Units: Dmnl)} \tag{A11}$$

$$\text{Effect of CPUE on boat entry} = \text{WITH LOOKUP (CPUE ratio, ([(0,0)-(1.2,0.2)], (0,0), (0.35,0.05), (1.12,0.1))) (Units: Dmnl)} \tag{A12}$$

$$\text{Effect of Density} = \text{WITH LOOKUP (Time/Initial year, ([(1,0)-(1.035,60)], (1,33.6842), (1.0009,34.4737), (1.00206,43.9474), (1.00226,42.3684), (1.00294,26.3158), (1.00579,36.5789), (1.00634,11.5789), (1.00765,9.21053), (1.00974,16.0526), (1.01105,15), (1.01116,16.3158), (1.01248,10.5263), (1.01335,6.84211), (1.01467,10), (1.01566,11.3158), (1.01675,8.42105), (1.01972,7.36842), (1.02136,6.05263), (1.02245,5), (1.02366,5.52632), (1.02531,5.52632), (1.0264,4.47368), (1.02805,5), (1.02966,5), (1.03259,3.94737), (1.03473,4.47368))) (Units: 1)} \tag{A13}$$

$$\text{"Effect of feed (plankton) on fish density"} = \text{WITH LOOKUP (Feed requirements met, ([(0,0.8)-(3,2)], (0,0.95), (1,1), (3,1.05))) (Units: Dmnl)} \tag{A14}$$

$$\text{Effect of profitability on boat acquire} = \text{WITH LOOKUP (Expected profitability, ([(0,0)-(1.5,2)], (0,0), (0.142202,0.587719), (0.220183,0.938596), (0.633027,1.02632), (0.802752,1.05263), (0.857798,1.24561), (0.981651,1.32456), (1.48624,1.11404))) (Units: Dmnl)} \tag{A15}$$

$$\text{Effect of Relative IM Adult} = \text{WITH LOOKUP (Relative IM Adult, ([(0,0)-(3,2)], (0,2), (0.3,1.7), (0.623853,1.35088), (1,1), (1.34862,0.684211), (1.84404,0.412281), (2.43119,0.219298), (3,0.1))) (Units: Dmnl)} \tag{A16}$$

$$\text{Effect of Relative Population on Density} = \text{WITH LOOKUP (Relative IM Adult, ([(0,0)-(3,2)], (0,0), (0.211009,0.614035), (0.66055,1.20175), (1.3945,1.66667), (2.22018,1.88596), (3,2))) (Units: Dmnl)} \tag{A17}$$

$$\text{"Effect of sea surface temperature (SST) on plankton biomass"} = \text{WITH LOOKUP (CLIMATE CHANGE, ([(20,0.5)-(40,1)], (20,1), (40,0.5))) (Units: Dmnl)} \tag{A18}$$

$$\text{Effects of Fish Density on Regeneration = WITH LOOKUP (Relative Density, ([(0,-2)-(2.2,6)],}$$
$$(0.0278287,-1.41579), (0.363303,0.140351), (0.552294,0.315789), (0.766972,0.45614),$$
$$(0.914985,0.385965), (1.06972,0.421053), (1.137,0.77193), (1.20428,1.15789), (1.48685,4.91228),$$
$$(1.60795,5.29825), (1.74251,5.40351), (1.97125,5.57895), (2.14495,5.61404)))) \text{ (Units: Dmnl)} \tag{A19}$$

$$\text{"Enter/leave delay"} = 1 \text{ (Units: Year)} \tag{A20}$$

$$\text{Entering or leaving fleet} = \text{SMOOTH(New boat rate, "Enter/leave delay")} \times \text{Effect of CPUE on boat entry}$$
$$\times \text{ Effect of profitability on boat acquire (Units: boat/Year)} \tag{A21}$$

$$\text{Expected profitability} = \text{(Revenue-Total operation cost)/Revenue (Units: 1)} \tag{A22}$$

$$\text{Feed requirements met} = \text{Actual phytoplankton biomass/Food demand (Units: 1)} \tag{A23}$$

$$\text{Female fraction} = 0.5 \text{ (Units: Dmnl)} \tag{A24}$$

$$\text{"Fishing mortality (catch)"} = \text{IM Adult} \times \text{Effect of Density} \times \text{Indicated catch fraction (Units: fish/Year)} \tag{A25}$$

$$\text{Food demand} = \text{IM Adult} \times \text{Normal phytoplankton requirement per fish per year (Units: ton/Year)} \tag{A26}$$

$$\text{Gear efficiency} = 0.5 \text{ (Units: Dmnl)} + \text{SMOOTH(STEP}(-0.25,2012), 4) \tag{A27}$$

$$\text{Growth fraction} = 0.08 \text{ (Units: Dmnl/Year)} \tag{A28}$$

$$\text{Growth rate} = \text{IM Juvenile/Growth Time (Units: fish/Year)} \tag{A29}$$

$$\text{Growth Time} = 0.5 \text{ (Units: Year)} \tag{A30}$$

$$\text{IM Adult} = \text{INTEG (Growth rate} - \text{"Fishing mortality (catch)"} - \text{Natural mortality, 60,000) (Units: fish)} \tag{A31}$$

$$\text{IM Juvenile} = \text{INTEG (Spawning-Growth rate, 30,000) (Units: fish)}$$
$$\text{fecundity in Indian mackerel ranges between 39,600 to 73,781}$$
$$\text{eggs} -> \text{Early developmental stages of the Indian mackerel}$$
$$\text{Rastrelliger kanagurta (Cuvier) along the Kerala-Mangalore}$$
$$\text{coast of southeastern Arabian Sea G. Sree Renjima, V. N.}$$
$$\text{Sanjeevan*, B. R. Smitha, C. B. Lalithambika Devi and M.}$$
$$\text{Sudhakar. Centre for Marine Living Resources and Ecology, Kochi, Kerala, India.} \tag{A32}$$

$$\text{Indicated boat} = \text{Total number of boats} \times \text{Gear efficiency (Units: boat)} \tag{A33}$$

$$\text{Indicated catch fraction} = \text{Normal catch fraction} \times \text{Effect of boat on catch (Units: 1/Year)} \tag{A34}$$

$$\text{Indicated Density} = \text{Normal Density} \times \text{Effect of Relative Population on Density} \times \text{"Effect of feed}$$
$$\text{(plankton) on fish density" (Units: 1)} \tag{A35}$$

$$\text{Initial year} = 1980 \text{ (Units: Year)} \tag{A36}$$

$$\text{Natural mortality} = \text{IM Adult} \times \text{Natural mortality fraction (Units: fish/Year)} \tag{A37}$$

$$\text{Natural mortality fraction} = 0.02 \times \text{(1-Climate change on lifespan) (Units: 1/Year)} \tag{A38}$$

$$\text{New boat rate} = 7500 \text{ (Units: boat/Year)} - \text{STEP(5625,2012)} \tag{A39}$$

$$\text{Normal catch fraction} = 1 \text{ (Units: 1/Year)} \tag{A40}$$

$$\text{Normal CPUE} = 12 \text{ (Units: fish/boat/Year)} \tag{A41}$$

$$\text{Normal Density} = 1 \text{ (Units: Dmnl)} \tag{A42}$$

$$\text{Normal phytoplankton requirement per fish per year} = 0.18 \text{ (Units: ton/fish/Year)}$$
$$\text{The volume of stomach ranged from 0.35 mL to 0.91 mL. The weight}$$
$$\text{of food consumed varied from 0.1 g to 1.2 g. Based on this fact,}$$
$$\text{we can assume that we know 39.9\% of the food consume is}$$
$$\text{phytoplankton, therefore taking the size of the stomach can}$$
$$\text{contain max will be 1.2 g} \times 39.9\% = 0.5 \text{ g of Phytoplankton/fish/year} \tag{A43}$$

$$\text{Normal Spawning Time} = 2 \text{ (Units: Year)} \tag{A44}$$

$$\text{Price} = \text{INTEG (Change in price, 213) (Units: USD/fish)} \tag{A45}$$

$$\text{Reference boat} = 2661 \text{ (Units: boat)} \tag{A46}$$

$$\text{Reference Density} = 1.416 \text{ (Units: Dmnl)} \tag{A47}$$

$$\begin{gathered}\text{Reference IM Adult} = 60{,}000 \text{ (Units: fish)}\\\text{initial value of stock IM adult (1980)}\end{gathered} \tag{A48}$$

$$\text{Reference phytoplankton biomass} = 10{,}800 \text{ (Units: ton/Year)} \tag{A49}$$

$$\text{Relative boat} = \text{Indicated boat/Reference boat (Units: 1)} \tag{A50}$$

$$\text{Relative Density} = \text{Indicated Density/Reference Density (Units: 1)} \tag{A51}$$

$$\text{Relative IM Adult} = \text{IM Adult/Reference IM Adult (Units: 1)} \tag{A52}$$

$$\text{Revenue} = \text{Price} \times \text{``Fishing mortality (catch)'' (Units: USD/Year)} \tag{A53}$$

$$\text{Spawning} = \text{Female fraction} \times \text{Batch per IM Adult} \times \text{IM Adult/Spawning time (Units: fish/Year)} \tag{A54}$$

$$\text{Spawning time} = \text{Normal Spawning Time} \times \text{Effects of Fish Density on Regeneration (Units: Year)} \tag{A55}$$

$$\text{Total number of boats} = \text{INTEG (Entering or leaving fleet-Boat discard rate, 5323) (Units: boat)} \tag{A56}$$

$$\text{Total operation cost} = \text{Cost per boat} \times \text{Total number of boats (Units: USD/Year)} \tag{A57}$$

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
