# Peer review of "A System Dynamics Approach for Improved Management of the Indian Mackerel Fishery in Peninsular Malaysia"

_sustainability, doi:10.3390/su142114190_

Round 1

Reviewer 1 Report

I have reviewed the paper based on my expertise in the field of fisheries anthropology, thus my comments will not be relevant for the modelling methods part. 

The paper is well written and interesting for fisheries managers, and should be published after some revisions. The main comments I have are the following: 

- The main findings and recommendations are fine targeted at fisheries managers as an audience. As an academic reader I am however left wondering about the theoretical contribution of the paper. Was there something from this modelling exercise that could contribute theoretically or practically to modelling theory and practice? A discussion on the application of this kind of modelling to management and what it can teach us about fish stock modelling would raise the paper additionally. Importantly, I find a critical perspective on the impacts of some of the policy scenarios for various groups of fishers lacking. What would the consequence of a rather radical reduction in number of fishing vessels and fisher mean for the economy and livelihoods of Malaysian fishers? Would they turn to other fisheries, thus over-exploiting other stocks if they are cut off from participation in the IM fishery? What are the limitations of such a model for providing advice to fisheries managers, if managers are not provided with estimates of social consequences from the model? 

- The system dynamics model is utilised based only on economic, biological and management models of the system. My main concern is the lack of reflection on social consequences of the policy scenarios as mentioned above. Another concern is however the lack of nuanced analysis of the impacts of reduction of vessel fleet in terms of the composition and catch patterns of the different parts of the fleet. In the end the paper importantly points to a reduction in trawlers and purse seiners. However there is little information on the composition of the vessel fleet from the outset. How much of the fleet consists of trawlers, and what about small-scale fishers? Some information in the beginning of the paper to anchor analysis of the implications of the reduction of fishing effort for different parts of the fleet would contribute to the analysis of the system as such. 

- The assumption for the model in the methods part of the paper states that 6. gear efficiency is treated as constant and assumed at 5 %. However, there is quite a lot of discussion about gear efficiency in the results and discussion part which runs counter to this assumption. I am also not convinced that the authors have enough information about what gear efficency in the IM fishery consitutes. Gear efficiency is not only about the performance of the vessel but goes hand in hand with what fishers can afford. The effectiveness of the gear might also very well increase when the number of boats are reduced as fishers can invest more due to increased catchers and increasing income. The authors should refine their analysis of gear efficiency in the model and explain what kind of gear is implied - purse seiners or trawl, or conventional gear? Also there is no description of the spatial dimension of the fishery. Will reducing number of vessels offshore impact the system differently from reduction of vessels in inshore areas? And where are the spawning areas located which should be protected from fisheries as implied in the latter part of the paper? Including a map of the area would be helpful. 

- The model at one place in the paper is explained as temporally extending over 33 years from 1980, however later in the paper the year 2016 is used which means the extent of the temporal dimension is 36 years. This should be corrected. 

- There is a general confusion with past and present tense activities in the paper. Was the model run already in methods part? If not, stick to present or future tense, as describing reductions and increases as having already happened in the methods part is confusing. 

- The models and illustrations themselves are nicely done, although figure 5 can be simplified. Note that as most readers will print in black and white, the coloured codes will have no meaning for readers. 

All in all, after revising these rather minor points and lifting up the theoretical and critical reflections of applying this method as advice for fisheries management will improve the paper further. I suggest starting and ending the paper with reflections on discussions about providing advice to fisheries management based on system dynamics models imply, and providing your own reflections based on the exercise presented in the paper. If this is part of a PhD thesis, those reflections definitely should be addressed in the synthesis part of the thesis. 

Author Response

Thank you Reviewer 1 for your extremely useful comments and suggestions.  Please find the attachment of my reply and justifications to both your comments and suggestions.

Thank you.

Reviewer 2 Report

This manuscript described a system dynamic approach for improved management of the Indian Mackerel Fishery in Peninsular Malaysia. The paper is interesting but the present version is far from publishable. This is because a quantitative comparison between the simulated results and historical data was not being investigated using the statistical method (such as the root-mean-square (RMS) errors, relative errors (RE), the predictive skill of the model).  In addition, English is very poor. After revising the manuscript as per the following suggestions and checking English by native English speaker, the author can resubmit to “Sustainability”.

Author Response

(The authors gave the same response as above.)

Reviewer 3 Report

1. The work adopted the systems dynamics approach to study resource management in Indian Mackerel (IM) in Malaysia. This approach can also provide references for other researchers to study the stock of fish species or improve systems dynamics assessment methods.

2. It is helpful for the researcher to reference the model benefits from the current management and data availability. This research paper may serve as a valuable reference for elsewhere in the world

3. References cite the same format for references. For example, 

     3. Asmala, A., Abd Rahman, M.A., Fadhli, A., Mustafa, M. and Khiruddin, A.              (2014). Euphotic depth zone variation in Peninsular Malaysia              maritime. Applied Mathematical Sciences, Vol. 8 (2014), no 68, pp 3375-3383. 

      4. Bay of Bengal Large Marine Ecosystem Project (BOBLME) (2012). Report  of the IM Working Group Meeting; 28-29 may 2012, Colombo, Sri Lanka. BOBLME-2012-Ecology-05.

Author Response

Thank you Reviewer 3 for your extremely useful comments and suggestions. Please find the attachment of my reply and justifications to both your comments and suggestions.

Thank you.

Reviewer 4 Report

This submission lacks proper line numbering, and as such, it is very difficult to specify detailed comments in this round. I suggest the authors consider the following comments carefully and revise the manuscript before resubmitting the ms with line numbers added.

Some general comments: 

1, The method section is not sufficient. Some descriptions of the model are difficult to follow. The equations for each transition and the meaning of each state variable should be specified either in a table or in a supplement?

2, Add a new section about how the model was calibrated. This step probably took the majority of the time, and it should be properly documented.

3, The discussion section is not sufficient. How to properly manage open access fisheries is an open problem. There are many ways to solve the problem. Restricting access may be a solution for this fishery, and to support this view, the authors need to analyze/discuss the feasibility of implementing/enforcing those restrictions from different perspectives. Otherwise, there is little practical/theoretical use in the model because we know the tragedy of commons leads to overexploitation in general.

Author Response

Thank you Reviewer 4 for your extremely useful comments and suggestions. Please find the attachment of my reply and justifications to both your comments and suggestions.

Thank you.

Round 2

Reviewer 2 Report

Check the statistical analysis.

Author Response

The authors thank the reviewer for the endorsement of the overall report as well as his/her comment.

Enclosed please find my response in regards to the comment made by the Reviewer 2.

Thank you.

Reviewer 4 Report

I am afraid I can not give a more favorable review as there are too many holes in the methods section. The authors should try to provide more details on the model specification. Integrating biological and economic dynamics is a BIG endeavor. Models of this scale usually have pages of equations and input parameters. I am afraid without seeing the details of the model and the fitting process, I am not confident about the results of this study, so I have not checked beyond the methods section.

See detailed comments below:

Line 123: I understand that this study is based on a model implemented using commercial software. While I have no experience with this particular software used by the authors, I have used other deterministic simulation software before like STELLA and MatLab. Typically, these simulations are heavily influenced by model assumptions and fitting methods, and model calibration and specification usually constitute a large part, if not the main part, of the research. But in this ms, the authors are very vague about these aspects. If some build-in procedures of the software were used, a relevant reference or link to its specification should be included. 

I need to say that I am not familiar with this piece of software. I might be wrong or missing something here. If Vensim is different from the more traditional simulation software like those mentioned before, the authors should include a sentence or two to describe the difference as most readers would have the same questions as well.

Please check the citation format of the journal and revise the citations throughout the ms.

line 113-114: The meaning of this sentence is not clear to me.

line 255: missing parenthesis

line 268, 273, 278: These equations don't make sense. First of all, the apparent unit of the variables in the equations is not compatible. All of these equations should be formulated as integrals.

Author Response

The authors thank the reviewer for his/her useful comments and suggestions.  

Enclosed please find the authors' response to the comments made.

Thank you.

Round 3

Reviewer 4 Report

Minor things like the format of the citations and references that need to be corrected. For example, the author-year style was used; reference 49 was added but never cited in the text. 

Author Response

Dear Reviewer 4,

Thank you for the comments made. I have edited the references accordingly and refined some of the sentences.